# Self-Replicating Alphaviruses: From Pathogens to Therapeutic Agents

**DOI:** 10.3390/v16111762

**Published:** 2024-11-12

**Authors:** Kenneth Lundstrom

**Affiliations:** PanTherapeutics, CH1095 Lutry, Switzerland; lundstromkenneth@gmail.com

**Keywords:** alphaviruses, viral particles, RNA replicons, DNA delivery, infectious diseases, cancer, vaccines, neurological disorders, therapy, clinical trials

## Abstract

Alphaviruses are known for being model viruses for studying cellular functions related to viral infections but also for causing epidemics in different parts of the world. More recently, alphavirus-based expression systems have demonstrated efficacy as vaccines against infectious diseases and as therapeutic applications for different cancers. Point mutations in the non-structural alphaviral replicase genes have generated enhanced transgene expression and created temperature-sensitive expression vectors. The recently engineered trans-amplifying RNA system can provide higher translational efficiency and eliminate interference with cellular translation. The self-replicating feature of alphaviruses has provided the advantage of extremely high transgene expression of vaccine-related antigens and therapeutic anti-tumor and immunostimulatory genes, which has also permitted significantly reduced doses for prophylactic and therapeutic applications, potentially reducing adverse events. Furthermore, alphaviruses have shown favorable flexibility as they can be delivered as recombinant viral particles, RNA replicons, or DNA-replicon-based plasmids. In the context of infectious diseases, robust immune responses against the surface proteins of target agents have been observed along with protection against challenges with lethal doses of infectious agents in rodents and primates. Similarly, the expression of anti-tumor genes and immunostimulatory genes from alphavirus vectors has provided tumor growth inhibition, tumor regression, and cures in animal cancer models. Moreover, protection against tumor challenges has been observed. In clinical settings, patient benefits have been reported. Alphaviruses have also been considered for the treatment of neurological disorders due to their neurotrophic preference.

## 1. Introduction

Alphaviruses are single-stranded RNA (ssRNA) viruses with a genome of positive polarity [1]. They are enveloped viruses possessing a structure consisting of trimeric spike proteins. Already in the 1980s, the life cycle of alphaviruses was described, including the major steps of host cell recognition, RNA delivery to host cells, RNA replication involving the viral replicase complex, translation of viral structural proteins, nucleocapsid formation, post-translational modifications, transport to the plasma membrane, and, finally, the release of mature viral particles by budding [1]. Alphaviruses have commonly been divided into Old and New World viruses. The most commonly investigated alphaviruses are the Semliki Forest virus (SFV) [2], Sindbis virus (SIN) [3], and Venezuelan equine encephalitis virus (VEE) [4]. Typically, the SFV and SIN belong to Old World viruses, while the VEE and eastern equine encephalitis virus (EEE) represent New World viruses [5].

The pathogenicity of alphaviruses is well documented causing febrile illness, arthritis and in severe cases encephalitis. Several epidemics have been associated with alphaviruses in different parts of the world [5].

The breakthrough in genetic engineering technologies made it possible to develop efficient expression systems for alphaviruses, especially for SFV [6], SIN [7], and VEE [8]. This development made great contributions to gene delivery both in vitro and in vivo when applied to prophylactic and therapeutic interventions in animal models and humans in clinical trials. Moreover, the neurovirulent nature of alphaviruses has made them attractive as delivery tools in basic neuroscience and potentially for therapeutic applications in the fields of neurodegeneration and neurological disorders [9].

In this review, the application of alphaviruses in basic research on the viral life cycle and cellular functions related to post-translational modifications and the transport of cellular and viral proteins is summarized. The pathogenicity of alphaviruses resulting in epidemics is also presented. Moreover, the engineering of various alphaviral vectors, attenuated alphavirus strains, and the identification of natural or engineered oncolytic alphaviruses are described. Additionally, vector and expression system developments are discussed. Finally, prophylactic and therapeutic interventions using alphaviruses are described.

## 2. The Alphavirus Life Cycle

The life cycle of alphaviruses has been investigated since the 1980s as a model for RNA viruses [10] (Figure 1). In addition to determining the crystal structure of alphavirus particles [11], the virus–host interaction including host cell recognition [12], virus entry [13], RNA replication [14], nucleocapsid assembly [15], post-translational modifications and transport of viral glycoproteins [16], and the release of mature viral particles [17] have been investigated and described in detail [1]. Briefly, host cell recognition has been identified as mediated by many different types of receptors explaining the broad host range including insects, vertebrates, and amphibians. It has also been demonstrated that, for example, different neurovirulent alphaviruses use different receptors [17]. Laminin [18] and heparan sulfate [19] have been identified as common receptors on mammalian cells for alphaviruses. The entry of viral particles involves fusion with the cellular membrane [20], the delivery of nucleocapsids to the cytoplasm [21], and the release of the RNA genome [22]. Alphavirus entry has also been described as occurring through endocytosis in clathrin-coated vesicles followed by transfer to endosomes [23,24]. Furthermore, it has been demonstrated that a low pH in the endosomes is essential, but voltage potential is also of importance [25]. Once the RNA genome is released, RNA replicase complexes (RCs), composed of the nonstructural proteins (nsP1–4), are assembled in modified endosomal and lysosomal membranes [26]. The RNA RCs remain stable throughout the infection cycle; moreover, a negative-strand RNA template is used for extensive viral RNA replication in host cells [27]. Due to the strong subgenomic promoter, alphavirus structural proteins are expressed at high levels leading to the assembly of RNA into C proteins, forming nucleocapsids [15]. Simultaneously, the viral envelope proteins are glycosylated and subjected to post-translational modifications in the endoplasmic reticulum (ER) and the Golgi [16]. The process of viral glycoprotein folding is initiated immediately after entry into the ER and requires molecular chaperons, folding enzymes, and the formation of disulfide bonds [28]. Subsequently, the envelope proteins are transported through the Golgi complex to the cell membrane [29]. Finally, during the maturation of alphaviruses, nucleocapsids are surrounded by envelope proteins leading to the release of mature alphavirus particles by budding [30]. A multitude of findings have not only substantially enhanced our understanding of the molecular and cellular aspects of alphaviruses, but have also enhanced our knowledge of cellular biology in general. The information on the pathogenesis and epidemiology of alphaviruses has also been enriched as described below.

## 3. Pathogenesis and Epidemiology of Alphaviruses

Different alphaviruses have been identified as pathogens in both animals and humans [31]. It is well-documented that several epidemics have been caused by alphaviruses. For example, VEE has been responsible for outbreaks in South America resulting in a few fatal cases in horses and fever epidemics in humans [17]. Moreover, SFV and SIN have been associated with outbreaks in Africa [32,33] and painful polyarthritis in Northern Europe [34]. Chikungunya virus (CHIKV) has caused epidemics in the Republic of Congo [35], the island of Reunion [36], and more recently in Brazil [37]. Taking this information into account, avirulent and attenuated alphavirus strains have been used for the engineering of expression systems.

Pathogenic effects have been related to host cell killing induced by the apoptosis of alphaviruses [38]. It has been shown that apoptosis is associated with a loss of membrane potential in mitochondria followed by the activation of caspases-3, -8, and -9 [39]. Another issue relates to the shut-off of the host cell protein synthesis by alphaviruses in infected cells [40]. Additionally, it has been demonstrated that nsP2 and nsP3, which are associated with the shut-off, specifically suppress antiviral pathways [40]. The pathogenicity of alphaviruses, especially related to their use as expression systems, has been addressed by engineering less cytotoxic vectors as described below.

## 4. Alphavirus Vector Development

Expression systems have been engineered for different alphaviruses. The most commonly used alphavirus expression systems are based on SFV [6], SIN [7], and VEE [8]. Both replication-deficient and -competent expression systems have been engineered. Generally, replication-deficient alphavirus particles are obtained by transfecting mammalian host cells with in vitro-transcribed RNA from an expression vector containing the alphavirus nsP1–4 genes, and the gene of interest (GoI) and a helper vector carrying the alphavirus structural protein genes [6] (Figure 2A). Only RNA from the expression vector will be packaged in viral particles as the RNA packaging signal is located either in the nsP1 or nsP2 gene [41]. For this reason, the generated viral particles do not carry RNA coding for structural proteins, and new viral particles cannot be produced. However, these “suicide particles” can infect host cells and express the GoI. Alternatively, the GoI can be introduced into the alphavirus full-length genome either downstream of the replicase genes or the structural genes (Figure 2B). In this case, in vitro-transcribed RNA from a plasmid DNA template is transfected/electroporated into mammalian host cells for the production of replication-proficient recombinant particles. These particles provide the expression of the GoI in infected host cells, as well as the production of new virus progeny. The advantage of using replication-proficient alphaviruses is the potential spread to adjacent cells, especially in vivo, and the prolongation of transgene expression. On the other hand, safety issues must be considered to prevent any uncontrolled spread of the virus. To prevent replication-proficient particle production, a split helper vector system was engineered for SFV where the SFV capsid and envelope proteins were introduced into separate SFV helper vectors [42]. This procedure eliminated the production of replication-proficient SFV particles. The application of alphavirus-based expression systems is flexible as, in addition to recombinant particles, RNA replicons can also be used. In this case, in vitro-transcribed RNA from replication-deficient and -proficient expression vectors can be directly used for delivery. However, due to the single-stranded nature of the RNA, it is highly sensitive to degradation, which strongly affects the efficacy of delivery [43]. For this reason, RNA replicons have been encapsulated in lipid nanoparticles (LNPs), which prevent RNA degradation and improve delivery [44]. Alternatively, the direct use of alphavirus DNA replicons as plasmid vectors has been possible by replacing the SP6 or T7 RNA polymerase promoter with a CMV promoter [45] (Figure 2C).

Independent of whether expression systems based on viral particles, RNA replicons, or DNA plasmids are applied, the efficient RNA amplification in host cells combined with the strong 26S subgenomic promoter generates extremely high levels of transgene expression [46].

Despite the encouraging experience of using alphavirus expression systems during the last twenty years, additional improvements have been made. For example, introductions of mutations into the replicase genes of SFV [47] and SIN [48] vectors have resulted in less cytopathogenic vectors providing enhanced and prolonged transgene expression. Moreover, temperature-sensitive mutant vectors have proven useful for neuroscience studies, where expression is mainly neuron-specific at 37 °C while it is glial-specific at 33 °C in primary neurons and hippocampal slice cultures [49]. Another approach has been to introduce the translation enhancement signal of the alphavirus capsid protein into the expression vector, which has resulted in a 5–10-fold increase in transgene expression levels [50]. Moreover, a bipartite trans-amplifying RNA (taRNA) system has been engineered, where the replicase genes have been replaced by the GoI in the expression vector, and the replicase genes are provided from an optimized non-replicating RNA (nrRNA) vector in trans [51]. This also substantially reduces the size of the expression vector and has generated a 10–100-fold increase in transgene expression [51].

## 5. Prophylactics and Therapeutics

Applying alphavirus expression systems to vaccine and drug development has received much attention lately. As alphaviruses are known for their transient expression profile due to the relatively rapid degradation of their RNA, the application range is restricted mainly to acute diseases. Other options should be considered for chronic diseases. The main areas of interest are infectious diseases and different cancers, for which both prophylactic and therapeutic approaches have been taken. In the case of neurological disorders, the main area of applications of alphaviruses comprise the establishment of animal models for the initial evaluation of therapeutic interventions before conducting clinical trials on humans. As alphavirus-based prophylactic and therapeutic interventions have been reviewed in more detail elsewhere [52], selected examples are described below.

### 5.1. Infectious Diseases

In the context of infectious diseases, VEE particles expressing either the Lassa virus (LASV) glycoprotein (GPC) or nucleoprotein (NP) showed protection against LASV challenges in immunized guinea pigs [53] (Table 1). Moreover, the immunization of guinea pigs with VEE particles expressing either the Junin virus (JUNV) or Machupo virus (MACV) resulted in protection against challenges with JUNV and MACV, respectively [54]. In another approach targeting VEE, VEE DNA replicons expressing the VEE capsid gene protected immunized mice [55] and macaques [56] against challenges with VEE.

Prime-boost immunizations with VEE particles expressing pre-membrane proteins and envelope proteins (prME) from the Dengue virus (DENV), as well as a vaccine based on prME expression from a conventional DNA plasmid, have been conducted [57]. Although three doses of plasmid DNA (DDD) or VEE particles (VVV) reduced viremia rates in macaques, only the combination of two doses of plasmid DNA and one dose of VEE particles (DDV) provided complete protection against DENV challenges [57]. In the case of the Zika virus (ZIKV), VEE RNA replicons expressing the ZIKV prME encapsulated in nanocarrier lipids (NCLs) protected mice against ZIKV challenges after a single immunization with 10 ng of RNA [58]. SFV particles have been employed for the expression of prME and NS1 proteins from the Louping ill virus (LIV) [59]. SFV-LIV-prME/NS1 particles protected 100% of mice against challenges with LIV in contrast to only partial protection obtained for a commercial inactivated whole-virus vaccine. Moreover, SFV particles expressing the HIV-1 envelope protein (Env) showed superior antibody responses compared to mice immunized with a conventional plasmid DNA or recombinant Env protein [60]. VEE DNA replicons have also been used for the expression of Env gp160, showing similar humoral immune responses in mice to those obtained with a conventional plasmid DNA vaccine but with 10–100-fold lower doses [61]. Furthermore, VEE-Gag particles (AVX001) were subjected to a phase I trial, but due to vaccine instability, the trial was prematurely terminated [62].

Due to the seasonal influenza virus epidemics, alphavirus vectors have also been applied for influenza A virus (IFVA) vaccine development. For example, only 10 μg of SFV RNA replicons expressing the IFVA HA gene protected 90% of immunized mice against IFVA challenges [63]. Moreover, approximately 100-fold lower doses of VEE RNA replicons were required for the protection of mice against IFVA challenges compared to synthetic mRNA [64]. As the bipartite taRNA system generates 10–100-fold enhanced transgene expression, 50 ng of SFV-HA taRNA was sufficient to protect mice against IFVA challenges [51].

In the context of COVID-19 vaccines, VEE RNA replicons containing the prefusion-stabilized SARS-CoV-2 spike (S) protein encapsulated in LNPs elicited strong immune responses in BALB/c mice [65]. The VEE-SARS-CoV-2 S LNPs (LNP-nCoV saRNA) showed good safety and tolerability in the first-in-human phase I clinical trial, and neutralization of SARS-CoV-2 was achieved in 15–48% of vaccinated individuals [66]. In a phase II study, the LNP-nCoV saRNA vaccine was administered at a prime dose of 1 μg followed by a booster vaccination with 10 μg, which resulted in higher seroconversion rates than in the phase I dose-ranging study [67]. Another RNA replicon-based vaccine candidate comprising the VEE-SARS-CoV-2 S RNA encapsulated in lipid inorganic nanoparticles (LION) elicited Th1-biased immune responses in macaques [68]. Furthermore, the safety, tolerability, and immunogenicity of the repRNA-COV2S-LION vaccine candidate were demonstrated in phase II/III [69], supporting the approval of emergency use authorization (EUA) of the vaccine in India [70]. In another approach, the LNP-encapsulated saRNA-based ARCT-154 vaccine was subjected to phase I–III trials, showing 56.6% efficacy against any COVID-19 and 95.3% efficacy against severe COVID-19 [71]. The ARCT-154 has been granted EUA in Japan [72]. The COVID-19 vaccine candidate EXG-5003, based on a temperature-sensitive controllable self-replicating RNA (c-srRNA) approach, is functional at 33 °C (skin temperature) and non-functional at 37 °C (core body temperature) [73]. Intradermal administration of naked c-srRNA carrying the SARS-CoV-2 S RBD elicited strong cellular immune responses but not humoral immunity. However, the EXG-5003 vaccine provided excellent priming capacity when combined with homologous or heterologous booster vaccinations [73]. Moreover, booster vaccinations with either BNT162b2 or mRNA-1273 mRNA vaccines after two primary doses of EXG-5003 elicited superior cellular immunity and long-term immunity compared to a homogenous immunization strategy in phase I/II [74].

Among non-viral diseases, anthrax caused by *Bacillus anthracis* has been targeted by expression of the *B. anthracis* protective antigen (PA) from SIN particles, which elicited PA-specific IgG and neutralizing antibodies and some protection in mice [75]. In another study, it was demonstrated that SFV DNA replicons expressing the PA gene elicited superior immune responses compared to those obtained after the administration of conventional plasmid DNA [66]. Furthermore, SFV-PA DNA replicons protected mice against challenges with the *B. anthracis* A16R strain [76]. Brucellosis has been targeted by the expression of the *Brucella abortus* translation initiation factor 3 (IF3) from SFV particles, which resulted in the protection of mice from challenges with *B. abortus* [77]. In the context of malaria, protection was achieved in mice after immunization with SIN particles expressing a cytotoxic T-lymphocyte (CTL) epitope (SYVPSAEQI) of the malaria parasite [78].

### 5.2. Cancers

Alphavirus vectors have been applied for both prophylactic and therapeutic evaluations for different cancers (Table 2). Due to the apoptotic activity of alphaviruses, SFV particles expressing the enhanced green fluorescent protein (EGFP) effectively killed human H358a non-small-cell lung cancer (NSCLC) cells in vitro and showed strong tumor regression in nu/nu mice with implanted H358a tumors [79]. Similarly, SFV-LacZ RNA elicited robust immune responses in mice [80]. Furthermore, a single administration of 0.1 μg of SFV-LacZ RNA protected mice from challenges with CT26 colon tumor cells [80].

Alphaviruses have targeted breast cancer in several studies. For instance, VEE particles expressing the extracellular domain (ECD) and the transmembrane (TM) domains of HER2 completely protected mice with HER2/neu tumors [81]. In phase I, the VEE-HER2-ECD/TM particles showed good tolerability, partial response (PR), and stable disease (SD) in stage IV HER2 overexpressing breast cancer patients [81]. Moreover, SIN DNA replicons expressing the HER2/neu gene inhibited tumor growth in mice with A2L2 breast tumor cells [82]. In another study, in comparison to conventional plasmid DNA delivery, SIN-HER2/neu DNA replicons required 80% less DNA to elicit similar immune responses and protect BALB/c mice against tumor challenges [83].

Human papillomavirus (HPV) has been the target for the prevention and therapy of cervical cancer. For example, immunization with VEE particles expressing the HPV E7 protein protected C57BL/6 mice against tumor challenges [84]. Furthermore, the immunization of mice carrying cervical tumors with SFV particles expressing the HPV E6-E7 fusion protein resulted in complete tumor eradication [85]. SFV-HPV E6-E7 particles (Vvax001) were administered to patients with cervical intraepithelial neoplasia in a phase I trial, which showed good safety and elicited specific immune responses in all 12 tested patients [86]. Cervical intraepithelial neoplasia grade 3 patients are currently subjected to a phase II trial with the Vvax001 vaccine candidate [87]. In another approach, VEE particles expressing the carcinoembryonic antigen (CEA) induced antigen-specific immunogenicity and prolonged the survival of patients with stages III and IV colorectal cancer in phase I [88].

Combination therapy with alphavirus vectors, antibodies, and drugs has turned out to be successful. For example, the combination of VEE particles with antagonist and agonist monoclonal antibodies has proven to be a promising alternative. Moreover, VEE particles expressing the tyrosine-related protein-2 (TRP-2) combined with the antagonist anti-CTL antigen-4 (CTLA-4) monoclonal antibody (mAb) generated complete regression of B16F10 melanoma tumors in 50% of immunized mice [89]. However, VEE-TRP-2 particles combined with the agonist anti-glucocorticoid-induced tumor necrosis factor receptor (GITR) mAb showed complete regression in 90% of mice [89]. In another study, the SFV DNA replicon co-expressing the vascular growth factor receptor-2 (VEGFR-2) and interleukin-12 (IL-12) was co-administered with the SFV DNA replicon co-expressing survivin and the β-hCG antigen to a B16 mouse melanoma model [90]. Co-administration resulted in superior tumor growth inhibition and survival compared to treatment with either SFV-VEGFR-2/IL-12 or SFV-survivin-β-hCG Ag DNA replicons alone. Moreover, although VEE particles expressing the prostate-specific membrane antigen (PSMA) elicited strong immune responses in mice [91], only weak PSMA-specific immunogenicity was discovered in castration-resistant metastatic prostate cancer (CRPC) patients in phase I [92].

Oncolytic alphaviruses have proven to be an attractive alternative for cancer therapy, due to their specific killing of tumor cells. For example, the oncolytic M1 alphavirus effectively killed triple-negative breast cancer (TNBC) cells [93]. Superior efficacy was observed after co-administration of M1 and doxorubicin, significantly reducing mouse tumor growth [86]. In another study, intravenous administration of the oncolytic SIN AR339 strain to mice with implanted cervical tumors resulted in substantial tumor regression [94].

### 5.3. Neurological Disorders 

Alphaviruses have demonstrated a strong neurotropism, making them potential delivery vectors for neurological disorders. As the demand for long-term expression is necessary for the therapeutic efficacy of chronic diseases, the transient nature of alphavirus expression might not be ideal, but alphavirus vectors have certainly demonstrated their feasibility in animal models for human diseases as described below and summarized in Table 3. For example, SFV particles expressing IL-10 have been evaluated in an experimental autoimmune encephalitis (EAE) model for multiple sclerosis (MS) [95]. Intranasal delivery of SFV-IL-10 particles resulted in therapeutic benefits in the EAE model in BALB/c mice. Moreover, SFV-based expression of the tissue inhibitor of metalloproteinase 2 (TIMP-2) inhibited EAE development in the mouse CNS [96]. In another study, SFV particles expressing the transforming growth factor β1 (TGF-β1) showed significant inhibition of EAE in BALB/c mice [97]. In another approach, the peripheral administration of SFV particles and myelin basic protein (MBP) elicited superior immune responses in the splenocytes of EAE-susceptible SJL mice compared to control mice [98].

## 6. Conclusions

This review aims to describe the research on the lifecycle of alphaviruses as models for viruses in general and the cellular aspects of protein translation, post-translational modifications, and transport. Moreover, as several alphaviruses have been associated with epidemics in domestic animals and humans, their spread and pathogenicity have been the targets of thorough investigations. Due to a breakthrough in molecular biology genetic engineering in the 1990s, applying alphavirus expression systems represented an attractive alternative for recombinant protein expression both in vitro and in vivo. In this context, necessary attention was given to reducing the cytopathogenic effects caused by alphaviruses on host cells. The application of attenuated and avirulent strains showed a substantial reduction in cytopathogenicity. However, extensive mutagenesis of the replicase genes provided extended survival of the host cells and enhanced recombinant protein expression levels. Moreover, temperature-sensitive mutant vectors have shown feasibility in neuroscience allowing neuron-specific or glial-specific expression by a simple switch of the culture temperature for primary neurons or hippocampal slice cultures. Additionally, temperature-sensitive vectors have also been applied for intradermal vaccinations against COVID-19. Although M1 has proven efficient for oncotherapy, SFV, SIN, and VEE systems have been most commonly used among alphavirus vectors. Recently, VEE-based approaches have been favored due to their potentially superior targeting of lymphoid tissues inducing strong immune responses after immunization.

Alphaviruses have been subjected to prophylactic and therapeutic interventions in the areas of infectious diseases, cancers, and neurological disorders, as summarized in Table 1, Table 2 and Table 3. One asset of alphaviruses is the flexibility to use recombinant viral particles, RNA replicons, and DNA replicon-based plasmids. Immunization with alphaviruses has provided protection against challenges with LASV, JUNV, MACV, DENV, ZIKV, IFVA, and SARS-CoV-2 in rodent models. Moreover, protection was also achieved against challenges with *B. anthracis*, *B. abortus*, and malaria in mice. Interestingly, in comparison to synthetic mRNA delivery, the administration of RNA replicons required 10–100-fold lower doses to obtain similar immune responses. Likewise, the doses used for DNA replicons are 10–100-fold lower compared to conventional plasmid DNA administration. These findings are important as RNA and DNA replicons can provide better efficacy and potentially fewer adverse events than their corresponding counterparts. In the context of clinical trials, compared to other viral vectors the number of human studies with alphaviruses are much fewer, which to some extent relates to the more recent engineering and lesser use of alphavirus expression systems. However, the recent COVID-19 pandemic has given a boost for alphavirus-based systems, especially those applying LNP-encapsulated RNA replicons. Although less advanced than vaccines based on synthetic mRNA, some encouraging results have been obtained in phase I and II trials, resulting in EUA being granted for the VEE RNA-LION vaccine in India.

In the context of cancer prevention and therapy, alphavirus vectors have elicited strong immune responses, tumor regression, and even total cure in rodent models. Moreover, clinical benefits have been obtained in clinical trials in cancer patients. Interestingly, the administration of alphavirus vectors expressing HPV E6 and E7 proteins resulted in immune responses in all tested cervical cancer patients in phase I, which bodes well for future prophylactic applications.

The transient nature of alphavirus-based expression systems has made them suitable for acute diseases such as infectious diseases and cancers. In the case of both prevention and therapy, short-term high-level expression is preferable, whereas in the case of chronic diseases it is advantageous to be able to supply long-term expression. However, in the context of neurological disorders, alphaviruses have proven useful for demonstrating the proof of concept in mouse EAE models for MS.

Overall, the success with using alphaviruses for prophylactic and therapeutic interventions has encouraged further investment in these delivery systems. Despite some setbacks in reproducing findings from preclinical studies in humans, additional vector engineering and dose optimization need to be addressed to further expand the application range of alphaviruses in the future.

## Figures and Tables

**Figure 1 viruses-16-01762-f001:**
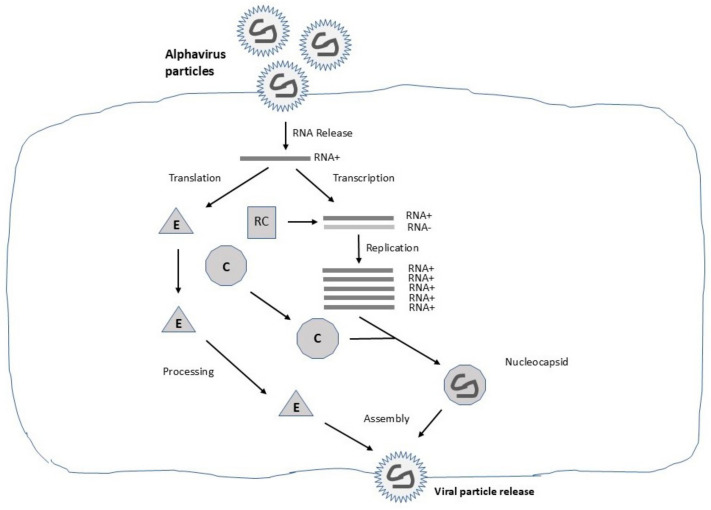
Schematic illustration of the life cycle of alphaviruses. Alphavirus particles deliver single-stranded RNA, where the ssRNA+ strand is transcribed and amplified. Among structural alphavirus proteins, the capsid (C) protein forms nucleocapsids (NCs) with the RNA genome, and the envelope (E) proteins are transported through the endoplasmic reticulum and Golgi for assembly with the NCs for release by budding of mature alphavirus particles.

**Figure 2 viruses-16-01762-f002:**
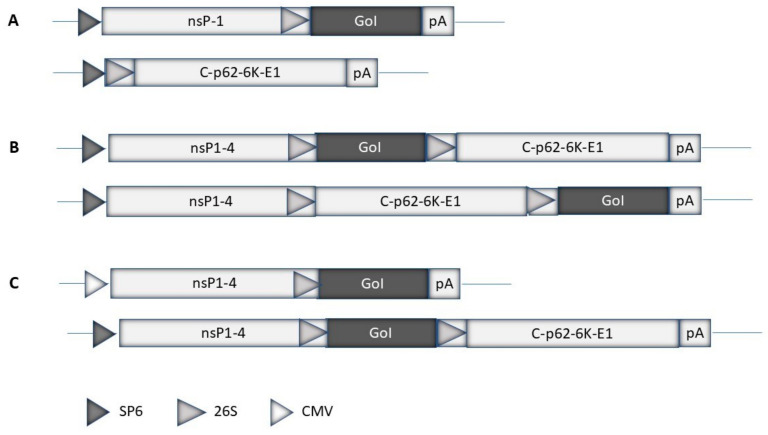
Schematic illustration of alphavirus expression systems. (**A**) Replication-deficient systems: The alphavirus expression vector contains the non-structural protein (nsP1-4) replicase genes and the gene of interest (GoI). The helper vector carries the structural protein genes (capsid (C)), p62, precursor of the E2 and E3 envelope protein, and E1 envelope protein). (**B**) Replication-proficient systems: The GoI can be inserted downstream of the nsP1-4 genes or, alternatively, downstream of the structural protein genes. (**C**) DNA replicon-based systems: Replication-deficient and -proficient vectors used for direct DNA plasmid transfer have been engineered by replacing the SP6 RNA polymerase promoter with the cytomegalovirus (CMV) promoter.

**Table 1 viruses-16-01762-t001:** Examples of prophylactic/therapeutic interventions with alphavirus vectors for infectious diseases.

Disease	Vector/Target Gene(s)	Findings
**Infections**		
LASV	VEE RP/LASV-GPC(NP)	Protection against LASV in guinea pigs [53]
JUNV	VEE RP/JUNV-GPC	Protection against JUNV in guinea pigs [54]
MACV	VEE RP/MACV-GPC	Protection against MACV in guinea pigs [54]
VEE	VEE DNA/VEE-Capsid	Protection in mice [55] and macaques [56]
DENV	VEE RP/DENV-prME + pDNA	Complete protection after prime-boost immunization in macaques [57]
ZIKV	VEE RNA/ZIKV-prME-NCLs	Protection against ZIKV challenges in mice [58]
LIV	VEE RP/LIV-prME/NS1	100% protection against LIV in mice [59]
HIV/AIDS	SFV RP/HIV-1 Env	Superior Ab response to pDNA and recombinant Env [60]
HIV/AIDS	VEE DNA/HIV-Env gp160	10–100-fold lower doses needed than pDNA [61]
HIV/AIDS	VEE RP/HIV-Gag	Phase I terminated due to vaccine instability [62]
IFVA	SFV RNA/IFVA-HA	90% protection with 10 μg RNA replicon in mice [63]
IFVA	VEE RNA/IFVA-HA	100-fold lower doses of RNA replicons [64]
IFVA	SFV taRNA/IFVA-HA	50 ng of taRNA sufficient for protection in mice [51]
COVID-19	VEE RNA-LNPs/SARS-CoV-2 S	Robust S-specific immune response in mice [65]
COVID-19	VEE RNA-LNPs/SARS-CoV-2 S	Good safety, immune responses in phase I [66]
COVID-19	VEE RNA-LNPs/SARS-CoV-2 S	Higher seroconversion rates in phase II than in phase I [67]
COVID-19	VEE RNA-LION/SARS-CoV-2 S	Th1-biased immunity in macaques [68]
COVID-19	VEE RNA-LION/SARS-CoV-2 S	Safe, tolerable, and immune responses in phase II/III [69]
COVID-19	VEE RNA-LION/SARS-CoV-2 S	EUA in India [70]
COVID-19	VEE RNA-LNPs/SARS-CoV-2 S	95.3% efficacy against severe COVID-19 in phase I–III [71]
COVID-19	VEE RNA-LNPs/SARS-CoV-2 S	EUA in Japan [72]
COVID-19	VEE c-srRNA/SARS-CoV-2 S RBD	Temperature-sensitive strong cellular immune response [73]
COVID-19	VEE c-srRNA/SARS-CoV-2 S RBD	Superior intradermal prime-boost immunization combined with approved mRNA vaccines in phase I/II [74]
Anthrax	SIN RP/*B. anthracis* PA	Immune responses, some protection in mice [75]
Anthrax	SFV RP/DNA/*B. anthracis* PA	Protection against *B. anthracis* A16R strain in mice [76]
Brucellosis	SFV RP/*B. abortus* IF3	Protection against *B. abortus* challenges in mice [77]
Malaria	SIN RP/*P. yoelii* CTL epitope	Protection against malaria in mice [78]

c-srRNA, controllable self-replicating RNA; DENV, Dengue virus; Env, envelope protein; EUA, emergency use authorization; GPC, glycoprotein; IF3, translation initiation factor 3; JUNV, Junin virus; LASV, Lassa virus; LION, lipid inorganic nanoparticles; LIV, Louping ill virus; LNPs, lipid nanoparticles; MACV, Machupo virus; NCLs, nanocarrier lipids; NP, nucleoprotein; pDNA, conventional plasmid DNA; PA, *B. anthracis* protective antigen; prME, pre-membrane and envelope proteins; RBD, receptor-binding domain; RP, recombinant particles; S, spike protein; SFV, Semliki Forest virus; SIN, Sindbis virus; taRNA, trans-amplifying RNA; VEE, Venezuelan equine encephalitis virus; ZIKV, Zika virus.

**Table 2 viruses-16-01762-t002:** Examples of prophylactic/therapeutic interventions with alphavirus vectors for cancers.

Cancer	Vector/Target Gene(s)	Findings
NSCLC	SFV RP/EGFP	Killing of H358a cells, tumor regression in mice [79]
Colon	SFV RNA/LacZ	Protection against CT26 tumor challenges in mice [80]
Breast	VEE RP/HER2 EDM/TM	Protection in mice, clinic benefits in phase I [81].
Breast	SIN DNA/HER2/neu	Protection in mice against A2L2 challenges [82]
Breast	SIN DNA/HER2/neu	Protection with 80% less DNA compared to pDNA [83]
Cervix	VEE RP/HPV E7	Protection against tumor challenges in mice [84]
Cervix	SFV RP/HPV E6-E7	Complete tumor eradication in mice [85]
Cervix	SFV RP/HPV E6-E7	Good safety, immunogenicity in all patients in phase I [86]
Cervix	SFV RP/HPV E6-E7	Phase II study in CIN3 patients in progress [87]
Colorectal	VEE RP/CEA	Safe, prolonged survival in phase I [88]
Melanoma	VEE RP + CTLA-4/TRP-2	Complete tumor regression in 50% of mice [89]
Melanoma	VEE RP + GITR/TRP-2	Complete tumor regression in 90% of mice [89]
Melanoma	SFV DNA/VEGFR-2/IL-12 + survivin/β-hGC	Superior tumor growth inhibition and survival after co-administration of SFV DNA replicons in mice [90]
Prostate	VEE RP/PSMA	Strong immune response in TRAMP mice [91]
Prostate	VEE RP/PSMA	Weak immunogenicity in CRPC patients in phase I [92]
Breast	Oncolytic M1 + doxorubicin	Strong TNBC tumor regression in mice [93]
Cervical	Oncolytic SIN AR339	Tumor regression in mice [94]

CEA, carcinoembryonic antigen; CIN3, cervical intraepithelial neoplasia grade 3; CTLA-4, anti-CTL antigen-4 mAb; EGFP, enhanced green fluorescent protein; GITR, glucocorticoid-induced tumor necrosis factor mAb; HPV, human papillomavirus; NSCLC; non-small-cell lung cancer; pDNA, conventional plasmid DNA; RP, recombinant particles; SFV, Semliki Forest virus; SIN, Sindbis virus; TNBC, triple-negative breast cancer; TRAMP, transgenic adenocarcinoma of the mouse prostate; TRP-2, tyrosine-related protein-2; VEE, Venezuelan equine encephalitis virus; VEGFR-2, vascular endothelial growth factor receptor-2.

**Table 3 viruses-16-01762-t003:** Examples of alphavirus-based disease models for neurological disorders.

Disease	Vector/Target Gene(s)	Findings
EAE	SFV RP/IL-10	Therapeutic benefits in EAE mouse model [95]
EAE	SFV RP/TIMP-2	Inhibition of EAE development in mice [96]
EAE	SFV RP/TGF-β1	Inhibition of EAE in BALB/c mice [97]
EAE	SFV RP + MBP	Superior immune responses in EAE-susceptible mice [98]

EAE, experimental autoimmune encephalitis; IL-10, interleukin-10; MBP, myelin basic protein; RP, recombinant particles; SFV, Semliki Forest virus; TGF-β1; transforming growth factor β1; TIMP-2, tissue inhibitor of metalloproteinase 2.

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
