# Peer review of "Self-Replicating Alphaviruses: From Pathogens to Therapeutic Agents"

_viruses, 2024, doi:10.3390/v16111762_

Round 1
Reviewer 1 Report
Comments and Suggestions for Authors
In this review, K. Lundstrom provides a comprehensive overview of the engineering and application of self-replicating alphaviruses as both prophylactic and therapeutic vectors across infectious diseases, cancer, and neurological disorders. The author discusses notable examples from each field, emphasizing recent advancements in developing safer and more effective vector systems.
The text is clear and easy to read.
Comments :
- Line 37/38 : “The alphaviruses have commonly been divided into old and new-world viruses.”
The author could maybe explain which Alphaviruses are in which group. Or distinguish them by “arthritogenic” versus “neurotropic” viruses?
- SINV and VEEV are named without the “V”, while other cited viruses are written with a V (SFV, DENV, LACV…).
- Figure 1 : “RC” is not defined (Replicase Complex, line 79?)
- Line 88 : “Finally, during the maturation of nucleocapsids surrounded by envelope proteins, mature alphavirus particles are released by budding [29].”
- Line 105 : “On the molecular level, some of the pathogenic effects have been related to the host cell killing induced by apoptosis of alphaviruses [37].”
This sentence is not clear, please reformulate
- Line 199 : “In contrast to a commercial inactivated whole virus, vaccine SFV-LIV-prME/NS1 particles protected 100% of mice against challenges with LIV.”
- Line 293 : Efficacy
- Line 314 : For example,
Author Response
In this review, K. Lundstrom provides a comprehensive overview of the engineering and application of self-replicating alphaviruses as both prophylactic and therapeutic vectors across infectious diseases, cancer, and neurological disorders. The author discusses notable examples from each field, emphasizing recent advancements in developing safer and more effective vector systems.
The text is clear and easy to read.
Comments :
- Line 37/38: “The alphaviruses have commonly been divided into old and new-world viruses.”
The author could maybe explain which Alphaviruses are in which group. Or distinguish them by “arthritogenic” versus “neurotropic” viruses?
Response: A brief text on Old and New World alphaviruses has been added.
- SINV and VEEV are named without the “V”, while other cited viruses are written with a V (SFV, DENV, LACV…).
Response: I acknowledge that V (for virus) is included in the abbreviations for SFV, DENV, LASV, etc. However, SIN and VEE are commonly used abbreviations in several publications.
- Figure 1: “RC” is not defined (Replicase Complex, line 79?)
Response: The “RC” abbreviation has been added.
- Line 88: “Finally, during the maturation of nucleocapsids surrounded by envelope proteins, mature alphavirus particles are released by budding [29].”
Response: The sentence has been modified accordingly.
- Line 105: “On the molecular level, some of the pathogenic effects have been related to the host cell killing induced by apoptosis of alphaviruses [37].”
This sentence is not clear, please reformulate
Response: The sentence has been modified accordingly.
- Line 199: “In contrast to a commercial inactivated whole virus, vaccine SFV-LIV-prME/NS1 particles protected 100% of mice against challenges with LIV.”
Response: The sentence has been modified accordingly
- Line 293: Efficacy
Response: The typo has been corrected.
- Line 314: For example,
Response: The correction has been made.
Reviewer 2 Report
Comments and Suggestions for Authors
The review is overall well-written, concise, and easy to read. It includes novel developments in the field of alphavirus vectors, making it interesting.
A few comments for the author:
line 20: In clinical settings, clinical benefits... (rephrase)
line 39: Semliki Forest virus (SFV) [2]. (should be a comma)
line 79: "The RNA replicase complexes remain stable throughout the infection cycle and are responsible for the extensive viral RNA replication in host cells from a negative-strand RNA template". In this sentence, I think the idea of the negative-strand RNA being an intermediate step also synthetized by the replicase complex is not clearly conveyed.
line 102: and Brazil [36] more recently. -- and more recently in Brazil.
line 105: On the molecular level, some of the pathogenic effects have been related to the host cell killing induced by apoptosis of alphaviruses -- On the molecular level, some of the pathogenic effects of alphaviruses have been related to...
Figure 1: the idea is clear but the aesthetics of the figure could be improved substantially.
line 110: nsP2 and nsP3 associated with the shut-off specifically suppress antiviral pathways -- nsP2 and nsP3, which are associated with the shut-off, specifically suppress antiviral pathways
Section 4: The different vectors are very well-explained, but I would include the possibility of using two helper vectors encoding C and Env proteins independently, which has been shown to be safer (at least to mention this in the text).
Figure 2: It is a bit difficult to identify the promoters quickly (in particular the 26S and the CMV). Maybe addition of colors in this figure could be useful.
Line 146: capsid (C)
line 154: the strong 26S subgenomic promoter
line 199: In contrast to a commercial inactivated whole virus vaccine,
line 239: (CTL)
lines 276 and 277: For example is repeated twice
line 293: eff8icacy
In the section 5.2, I understand why the author decided to mention at first the antitumor effect of SFV vectors expressing molecules that in principle have no therapeutic effect (eg EGFP or LacZ), but I was expecting to read more about vectors encoding therapeutic proteins such as IL12 or checkpoint inhibitors, even if the author wanted to keep this section brief.
line 314: For example,
line 340: Sensitivity to temperature has also been developed for other applications, not only neurosciences. For example for intradermal vaccination https://doi.org/10.1016/j.isci.2023.106335 (already tested in a clinical trial for COVID-19: EXG-5003)
line 358: LNP-encapsulated
line 366: maybe comment that a phase II that is ongoing (NCT06015854)
lines 376-377: further is repeated twice
Finally, and extra suggestion: It seems that the last part of the abstract and also the introduction are a bit disorganized; mentioning the developments and improvements without a very clear line of thought, and also mentioning sections of the review in a different order (eg. pathogenicity and epidemics after vector engineering). I suggest these parts could be improved.
Author Response
The review is overall well-written, concise, and easy to read. It includes novel developments in the field of alphavirus vectors, making it interesting.
A few comments for the author:
line 20: In clinical settings, clinical benefits... (rephrase)
Response: The sentence has been revised.
line 39: Semliki Forest virus (SFV) [2]. (should be a comma)
Response: The comma has been introduced.
line 79: "The RNA replicase complexes remain stable throughout the infection cycle and are responsible for the extensive viral RNA replication in host cells from a negative-strand RNA template". In this sentence, I think the idea of the negative-strand RNA being an intermediate step also synthetized by the replicase complex is not clearly conveyed.
Response: The sentence has been revised accordingly.
line 102: and Brazil [36] more recently. -- and more recently in Brazil.
Response: The sentence has been revised.
line 105: On the molecular level, some of the pathogenic effects have been related to the host cell killing induced by apoptosis of alphaviruses -- On the molecular level, some of the pathogenic effects of alphaviruses have been related to...
Response: The sentence has been revised.
Figure 1: the idea is clear but the aesthetics of the figure could be improved substantially.
Response: I respect the view of the reviewer, but I do not see how the aesthetics of the figure could/should be improved.
line 110: nsP2 and nsP3 associated with the shut-off specifically suppress antiviral pathways -- nsP2 and nsP3, which are associated with the shut-off, specifically suppress antiviral pathways
Response: The sentence has been modified accordingly
Section 4: The different vectors are very well-explained, but I would include the possibility of using two helper vectors encoding C and Env proteins independently, which has been shown to be safer (at least to mention this in the text).
Response: A short description of the SFV two-helper system has been added.
Figure 2: It is a bit difficult to identify the promoters quickly (in particular the 26S and the CMV). Maybe addition of colors in this figure could be useful.
Response: The three promoters are colored in different shades of grey. Perhaps slightly darker shades could be used for the SP6 and 26S promoters. Another solution is to use color, but that should be approved by the journal without any additional costs.
Line 146: capsid (C)
Response: The correction has been made.
line 154: the strong 26S subgenomic promoter
Response: The correction has been made.
line 199: In contrast to a commercial inactivated whole virus vaccine,
Response: The sentence has been revised.
line 239: (CTL)
Response: The correction has been made.
lines 276 and 277: For example is repeated twice
Response: The second “For example” has been replaced by “Moreover”
line 293: eff8icacy
Response: The correction has been made.
In the section 5.2, I understand why the author decided to mention at first the antitumor effect of SFV vectors expressing molecules that in principle have no therapeutic effect (eg EGFP or LacZ), but I was expecting to read more about vectors encoding therapeutic proteins such as IL12 or checkpoint inhibitors, even if the author wanted to keep this section brief.
Response: I do understand the point of the reviewer, but as mentioned in the text (L185-187) recent thorough reviews have covered the topic and it felt unnecessary to repeat the same findings here.
line 314: For example,
Response: The correction has been made.
line 340: Sensitivity to temperature has also been developed for other applications, not only neurosciences. For example for intradermal vaccination https://doi.org/10.1016/j.isci.2023.106335 (already tested in a clinical trial for COVID-19: EXG-5003)
Response: The temperature-sensitive c-srRNA-based intradermal administration system (including phase I/II evaluation) has been added to the text and Table 1.
line 358: LNP-encapsulated
Response: The correction has been made.
line 366: maybe comment that a phase II that is ongoing (NCT06015854)
Response: A sentence about the phase II trial has been added (+ reference 87).
lines 376-377: further is repeated twice
Response: The second “further” has been replaced by “additional”.
Finally, and extra suggestion: It seems that the last part of the abstract and also the introduction are a bit disorganized; mentioning the developments and improvements without a very clear line of thought, and also mentioning sections of the review in a different order (eg. pathogenicity and epidemics after vector engineering). I suggest these parts could be improved.
Response: The Abstract and the Introduction have been revised.
Reviewer 3 Report
Comments and Suggestions for Authors
Major points:
1. Please update Figure 1 to show the membrane-specific replication that is mentioned in the Figure Legend. Overall, the quality of this Figure could be improved to better clarity.
2. Lack of references for ARCT-154, an approved saRNA vaccine in Japan, diminishes the thoroughness of Table 1. There are several studies, the latest by Ho et al., 2024 Nat Comm, that summarizes clinical trials for this vaccine that should be included.
3. This sentence in the abstract needs to be modified or removed: "The recently engineered trans-amplifying RNA system has facilitated vector production and enhanced the expression capacity." For synthetic saRNA, there is no theoretical limit to expression capacity. The limitations are associated with manufacturing of long RNAs that are not optimized processes. Additional challenges with ratios of the replicase to the GOI with taRNA add complexity for product development that has limited their advancement. Based on this and lack of elaboration on taRNA, the above statement is inaccurate.
Minor point:
1. Discussion would benefit why VEE is the most commonly used vector vs other alphaviral vectors
2. One typo noticed on line 293 "eff8icacy"
Author Response
1. Please update Figure 1 to show the membrane-specific replication that is mentioned in the Figure Legend. Overall, the quality of this Figure could be improved to better clarity.
Response: I respectfully disagree with this comment. The point is to make the illustration as simple as possible and the introduction of membranes would make the figure too complicated. In the same way, neither the endoplasmic reticulum nor the Golgi apparatus is shown although mentioned in the text. In fact, the membrane-specific replication is not mentioned in the figure legend but in the text.
2. Lack of references for ARCT-154, an approved saRNA vaccine in Japan, diminishes the thoroughness of Table 1. There are several studies, the latest by Ho et al., 2024 Nat Comm, that summarizes clinical trials for this vaccine that should be included.
Response: The ARCT-154 vaccine candidate has been introduced into the text and added to Table 2 including its approval in Japan.
3. This sentence in the abstract needs to be modified or removed: "The recently engineered trans-amplifying RNA system has facilitated vector production and enhanced the expression capacity." For synthetic saRNA, there is no theoretical limit to expression capacity. The limitations are associated with manufacturing of long RNAs that are not optimized processes. Additional challenges with ratios of the replicase to the GOI with taRNA add complexity for product development that has limited their advancement. Based on this and lack of elaboration on taRNA, the above statement is inaccurate.
Response: The text has been revised accordingly.
Minor point:
1. Discussion would benefit why VEE is the most commonly used vector vs other alphaviral vectors
Response: A sentence has been added to the Conclusions section on the favorable use of VEE.
2. One typo noticed on line 293 "eff8icacy"
Response: The correction has been made.